# An Improved Instance Segmentation Method for Complex Elements of Farm UAV Aerial Survey Images

**DOI:** 10.3390/s24185990

**Published:** 2024-09-15

**Authors:** Feixiang Lv, Taihong Zhang, Yunjie Zhao, Zhixin Yao, Xinyu Cao

**Affiliations:** 1School of Computer and Information Engineering, Xinjiang Agricultural University, Urumqi 830052, China; 320223343@xjau.edu.cn (F.L.); 320192868@xjau.edu.cn (Z.Y.); 320213568@xjau.edu.cn (X.C.); 2Ministry of Education Engineering, Research Center for Intelligent Agriculture, Urumqi 830052, China; 3Xinjiang Agricultural Informatization Engineering Technology Research Center, Urumqi 830052, China

**Keywords:** image processing, instance segmentation, SparseInst, attention mechanism, farm aerial survey

## Abstract

Farm aerial survey layers can assist in unmanned farm operations, such as planning paths and early warnings. To address the inefficiencies and high costs associated with traditional layer construction, this study proposes a high-precision instance segmentation algorithm based on SparseInst. Considering the structural characteristics of farm elements, this study introduces a multi-scale attention module (MSA) that leverages the properties of atrous convolution to expand the sensory field. It enhances spatial and channel feature weights, effectively improving segmentation accuracy for large-scale and complex targets in the farm through three parallel dense connections. A bottom-up aggregation path is added to the feature pyramid fusion network, enhancing the model’s ability to perceive complex targets such as mechanized trails in farms. Coordinate attention blocks (CAs) are incorporated into the neck to capture richer contextual semantic information, enhancing farm aerial imagery scene recognition accuracy. To assess the proposed method, we compare it against existing mainstream object segmentation models, including the Mask R-CNN, Cascade–Mask, SOLOv2, and Condinst algorithms. The experimental results show that the improved model proposed in this study can be adapted to segment various complex targets in farms. The accuracy of the improved SparseInst model greatly exceeds that of Mask R-CNN and Cascade–Mask and is 10.8 and 12.8 percentage points better than the average accuracy of SOLOv2 and Condinst, respectively, with the smallest number of model parameters. The results show that the model can be used for real-time segmentation of targets under complex farm conditions.

## 1. Introduction

The construction of UAV aerial maps for farms has significant applications in crop monitoring, smart farm path planning [1], and farm machinery operations [2]. Particularly, it provides critical monitoring and decision-making support for unmanned farm machinery operations on large farms. When farm machines are manually operated, professionals are needed to drive large machinery, incurring labor costs. While driverless farm machines can avoid these issues, building a farm aerial survey layer is necessary to ensure safe autonomous operation. Deep learning technology enables precise recognition of farm images [3], allowing farm machinery to anticipate target locations within the farm layer, thereby avoiding potential risks. This capability enhances the accuracy of unmanned farm machinery decisions, improving both efficiency and safety. Consequently, the study of UAV farm image segmentation technology holds considerable practical significance [4].

Creating high-precision farm maps necessitates extensive detailed annotations. Traditional methods of layer construction rely on manual annotation, which is inefficient and prone to inaccuracies in boundary segmentation, resulting in significant labor and time expenditure. With the ongoing advancement of instance segmentation algorithms, both the accuracy and speed of segmentation have significantly improved. Instance segmentation methods [5,6], which differentiate targets and backgrounds at the pixel level, have thus become the standard for processing high-precision images. Compared to semantic segmentation, instance segmentation [7] offers the advantage of recognizing individual targets without confining objects to the same category. Unmanned farm machinery requires precise identification of all objects during operation, making instance segmentation highly suitable for constructing farm UAV aerial survey layers [8]. Unlike cityscapes and roads with regular geometries, farm scenes present unique challenges due to their natural features [9,10]. Farms typically consist of extensive arable land, mechanized paths, forest belts, and heathland, which feature minimal human intervention and exhibit complex characteristics, significant variations in scale among similar elements, and intricate edge details. These factors complicate pixel-level instance segmentation in farm imagery.

The rest of the paper is organized as follows. Section 2 presents research results on instance segmentation in the field of UAVs. Section 3 describes the process of collecting and constructing the farm aerial survey dataset. Section 4 details the SparseInst network and the improvement methods, including the multi-scale attention mechanism, aggregation path, and CA attention module, tailored to the characteristics of farm target structures. Section 5 analyzes the experimental results and includes the design and execution of comparison, ablation, and validation experiments on the HRSID. The conclusions are presented in Section 6.

## 2. Related Works

In recent years, extensive research has been conducted on the segmentation of remote sensing images. In 2017, Kaiming He et al. [11] introduced the Mask-RCNN instance segmentation network, which achieves pixel-level accuracy in image segmentation. Mask-RCNN operates as a two-stage network: it first detects candidate regions for object detection and subsequently segments instance masks based on these regions. Following Mask-RCNN, several networks such as Faster R-CNN [12], YOLACT [13], and PANet [14] have been developed, which build upon either candidate regions or anchor-based approaches for segmentation. Another category is single-stage segmentation networks that do not rely on anchors. These networks generate instance masks through aggregation and mask encoding rather than depending on anchor-based methods. Examples include CondInst [15], SOLO [16], SparseInst [17], and similar architectures. In the realm of remote sensing image segmentation, researchers have made substantial progress. For instance, Xiaocheng Zhou et al. [18] and colleagues achieved low-cost monitoring of young fir forests using UAV remote sensing images, with the measurement accuracy for individual saplings reaching 0.9968. Braga et al. [19] used the Mask R-CNN model to extract tree crowns from high-resolution WorldView-2 satellite imagery, achieving an F1 score of 0.86 for the extracted tree crowns. Jian et al. [20] designed a specialized CutPaste self-supervised method based on the Mask R-CNN network architecture to generate dense and reliable candidate anchors with the help of the region proposal network (RPN) network, enabling the segmentation and recognition of ships in remote sensing images. Yuanyuan Wang et al. [21] applied the fully convolutional network (FCN) semantic segmentation model to GaoFen-3 remote sensing images, completing the extraction and segmentation of water bodies, vegetation, and buildings. Beibei Xu et al. [10] proposed a new method based on the DeepLabv3plus network, combining the visible light color index with an instance segmentation method based on an encoder–decoder architecture. This approach achieves accurate detection and segmentation of densely distributed weeds and soybean crops, with a weed segmentation accuracy of 0.905. Yunpeng Wu et al. [22] developed a YOLARC framework for ranking potential safety hazards (PSHs) by utilizing UAVs to perform track inspections on high-speed railroads. YOLARC employs multiple convolutions with enlarged receptive fields to create a new backbone that can accurately predict regions with PSHs. Ran Zou et al. [23] evaluated the damage level of post-disaster buildings by integrating the mixed local channel attention (MLCA) mechanism into the YOLOv5-Seg backbone. The improved YOLOv5-Seg was three times more efficient than the Mask-R-CNN and YOLO V9-Seg models. Zhihao Guan et al. [24] developed an instance segmentation method (MaskSU R-CNN) based on the channel domain attention mechanism by incorporating the dual semantic attention (DSA) module into the ResNet backbone network. Using the fire image dataset FLAME, MaskSU R-CNN achieved 91.85% accuracy in early forest fire detection and segmentation. Huan Liu et al. [25] proposed a novel UAV remote sensing segmentation model called Clusterformer, which is an encoder–decoder structure, with the encoder consisting of a spatial channel feed-forward network (SC-FFN). The model is used to segment pine wilt disease (PWD), and its segmentation results on two PWD datasets are significantly better than those of existing state-of-the-art segmentation models.

Most studies in this area focus on target detection for segmentation, with a majority of segmented objects being relatively small. However, farm UAV images differ significantly from traditional datasets; for instance, a single piece of cultivated land in a farm image can encompass tens of thousands of pixels, and the narrow, complex structure of mechanized roads can cover the entire image. Using instance segmentation based on anchor frames often leads to issues such as high repetitiveness of anchors, excessively large anchor areas, excessive background features within anchors, and difficulties in target mask extraction. Therefore, this study adopts the SparseInst instance segmentation algorithm, which is based on pixel aggregation. This method utilizes instance activation maps to weigh and highlight relevant features, overcoming the limitations associated with anchor-based approaches. It proves to be advantageous for segmenting large-scale targets and complex features in farm images. The main contributions of this study are as follows.

In this study, the multi-scale attention module (MSA) is designed to address the challenges of target scale variations and irregular shapes in farm images. This module employs three atrous convolutions along with dense connectivity to extract multi-scale feature information, enabling it to capture a larger receptive field.

In this study, a bottom-up aggregation path is introduced to enhance the fusion of semantic information from low-level feature maps. This addition improves the model’s detection performance by more effectively integrating details from earlier layers with higher-level features.

Incorporating the CA attention mechanism into the pyramid fusion network significantly improves the convolutional neural network’s capability to capture long-range dependencies and extract detailed features.

## 3. Instance Dataset

The farm image data used in this study were collected from Huaxing Farm in Changji City, Xinjiang, China, Uygur Autonomous Region, using a DJI Mavic 3 UAV equipped with a visible light camera at a flight altitude of 100 m. The images were captured with a resolution of 5280 × 3956 pixels and a 24-bit JPG color depth. Instance segmentation labeling of the farm images was performed using the Labelme tool [26], resulting in a total of 1155 finely labeled images. The labeling mask is illustrated in Figure 1.

In order to maintain the target resolution of the images, scaling was not employed. However, due to the high computational demands associated with the original images, which have a large resolution [27], this experiment involves cropping the mask-labeled maps generated after labeling, along with the original images. A total of 10,395 images with a resolution of 1760 × 1319 were generated for each of the three image types—original, semantic labeled, and instance labeled—at a 1:9 ratio. The cropped mask maps were then restructured into a COCO [28] format dataset [29], as illustrated in Figure 2.

The farm dataset was split into training and test sets with a 4:1 ratio. The dataset was annotated with a total of eight categories, with the majority of farm elements being plotlands. After cropping at a 1:9 ratio, plotlands emerged as the most numerous category followed by forest belts and heathland. Outletpiles were distributed within the plots but were present at the smallest scale and in limited quantities, making them challenging to segment. The distribution of the farm categories is detailed in Table 1.

## 4. Methodology

### 4.1. SparseInst Instance Segmentation

Many instance segmentation networks combine target detection by adding a mask segmentation head beneath the generated anchor box to further segment instances. However, this approach is limited by the location of the anchor box, making it difficult to segment large targets. SparseInst [17] introduces a novel method using instance activation maps to extract feature weights, followed by a mask extraction branch to complete the segmentation. This avoids the limitations of traditional anchor-based mask extraction, making it well suited for instance segmentation of farm images. The method is innovative, achieves high accuracy, and uses a lightweight network architecture. The network structure is illustrated in Figure 3.

The SparseInst network constructs the instance activation mapping (IAM) segmentation decoder after the feature pyramid fusion network, consisting of an instance branch and a mask branch. First, the instance feature branch generates instance-aware weighted weights for the fused semantic features, which are then passed to the mask branch for precise instance segmentation. This design enhances the segmentation process by dynamically adapting to the features of each instance, improving accuracy, particularly for large and complex targets. This process involves four 3 × 3 convolutional layers, followed by a sigmoid activation function to produce feature probability values ranging from 0 to 1. The instance activation map is then derived by summing these values along the last dimension. This step can be expressed as follows. Given the input image feature $X\in R^{D\times (H\times W)}$, the instance activation map A is calculated as $A=F_{IAM}(X)\in R^{N\times (H\times W)}$, where A represents the sparse N-instance activation map and $F_{iam}(\cdot )$ denotes the sigmoid nonlinear activation function. Subsequently, a multiplication operation is performed with the reshaped features A, aggregating target information based on the activation map. Finally, three linear layers are applied for classification, target scoring, and generating the mask kernel. The mask branch then further refines the features using four 3 × 3 convolutions. The mask kernel obtained from the instance branch is matrix multiplied by the output from the mask branch to generate the predicted mask information. This process enhances the instance-aware features through instance-activated mapping, as shown in Equation (1).
(1)$$z=\bar{A}\cdot X^{T}\in R^{N\times D}$$
where $z={\left\{z_{i}\right\}}^{N}$ is the feature representation of *N* potential objects in the image and *A* is normalized for each instance map. The sparse instance-aware features ${\left\{z_{i}\right\}}^{N}$ are directly used for subsequent recognition and instance-level segmentation.

Due to farm elements’ scale variations and irregular shapes, this study introduces the MSA. This module employs three atrous convolutions with dense connections to perform multi-scale feature extraction. Using atrous convolutions allows for a larger receptive field and more efficient feature extraction. Additionally, a bottom-up aggregation path is incorporated into the neck network, facilitating comprehensive information transfer across different levels. The CA mechanism is designed to capture long-range dependencies and preserve detailed positional information. By importing this mechanism into the feature fusion network, important features can be captured more effectively. The improved network structure is illustrated in Figure 4.

### 4.2. Multi-Scale Attention Mechanism

To address the complex target features and large-scale variations in farm scenes, which traditional attention mechanisms struggle to handle, this study introduces an MSA mechanism. This mechanism is tailored to the characteristics of farm targets and uses parallel atrous convolutions for multi-scale semantic extraction, enhancing both channel and spatial semantic information. Although the channel attention (SENet) [31] can effectively find out the importance level between channels, the full connectivity layer used introduces new parameters that tend to cause overfitting. In this study, two 1 × 1 convolutions of the pyramid split attention (PSA) module in the EPSANet [32] network are used to replace the fully connected layer. The structure is shown in Figure 5, and this approach captures the dependencies between channels while effectively avoiding the complex process of dimensionality reduction and dimensionality upgrading and reduces the number of parameters, which enhances the transfer of information between channels.

Spatial features can be enhanced using the simple attention mechanism (Simam) [33] parameterless module, which deals with each position of the feature map through a pixel-by-pixel adaptive mechanism, which means that each pixel’s position can be independently adjusted with its weight. In contrast, the SENet mechanism involves a more complex structure and greater computational demands because it processes global information. Compared to the Simam module, feature enhancement can be more precisely controlled, resulting in improved expressiveness of the features. The Simam module’s simple structure and absence of parameters reduce the risk of overfitting. The Simam mechanism emulates the spatial inhibition behavior of neurons. It identifies a measure of linear separability between the target neuron and other neurons, with the energy function defined as shown in Equation (2).
(2)$$e_{t}(w_{t},b_{t},y,x_{i})={\left(y_{t}-\hat{t}\right)}^{2}+\frac{1}{M-1}\sum_{i=1}^{M-1}\;{\left(y_{o}-{\hat{x}}_{i}\right)}^{2}$$
where *t* and $x_{i}$ are the target neuron and other neurons in a single channel of the input feature, $X\in R^{C\times H\times W}$ is the index of the spatial dimension, and M=H×W is the number of neurons on that channel. $w_{t}$ and $b_{t}$ are the weights and biases of the transformation.

Despite the enhancement of semantic features through spatial and channel-wise attention, significant scale differences among targets in farm scenes persist. To address this, this study implements three parallel atrous convolutions for multi-scale feature extraction. A key advantage of atrous convolution is its ability to expand the receptive field [34] without increasing the number of parameters or computational cost. The receptive field is defined by the formula given in Equation (3).
(3)$$F_{i}=(F_{i-1}-1)\times stride+k$$
where $F_{i}$ denotes the receptive field of the *i* layer, $F_{i-1}$ denotes the receptive field of the i−1 layer, *stride* denotes the table step, and *k* denotes the convolutional kernel size. This is experiment number 3. Features at various scales are extracted by employing three different dilation rates of 1, 2, and 4. This multi-scale feature extraction capability allows the model to more effectively extract targets with significant scale variations. The structure is further optimized through dense connectivity with parallel connections [35]. As illustrated in Figure 6, the output of one atrous convolution [36] is used as the input for the subsequent atrous convolution. Additionally, the outputs of both the previous and current layers are concatenated along the channel dimension using the concat operation, which enhances the model’s expressive power.

The structure of the MSA network designed for this experiment is illustrated in Figure 7. The fully connected layer is replaced by two 1 × 1 convolutions to obtain channel semantic information weights, which are then matrix multiplied with the input features to produce the enhanced channel semantic information weighted map. Another branch applies three parallel atrous convolutions for large-scale feature extraction. However, this parallel design does not optimally utilize multi-scale semantic information and shares limitations similar to the feature pyramid network (FPN), often resulting in an incomplete representation of small-scale semantic information in the feature maps, leading to potential loss. To address this, a dense connection structure is introduced between the three atrous convolutions, as depicted in Figure 6. This design fuses small-scale and complex structural semantic information through channel concatenation, ensuring the retention of small-scale semantic information while extracting large-scale targets. Following a 1 × 1 dimensionality reduction, the output is summed with the channel information weighted map and the features enhanced by the Simam mechanism to generate the multi-scale semantic feature map. The results demonstrate that this multi-scale attention module (MSA) significantly enhances the extraction of large-scale target features within the farm environment. 

### 4.3. Feature Fusion Network

In the SparseInst neck network, features from the last three stages of ResNet are utilized with a top-down path for fusing features at different scales. Despite this multi-scale fusion, the combination of detailed and high-level information may still be insufficient for capturing underlying semantics. This is because low-level semantic information often contains richer details than high-level semantic information. Specifically, features of small targets may become increasingly abstracted and lost as semantic information is extracted through successive convolutional layers. In this study, a bottom-up aggregation path is incorporated into the path aggregation network (PAFPN) structure. This enhancement allows for a more effective fusion of semantic information from low-level feature maps, thereby improving the model’s detection performance. The revised network architecture is illustrated in Figure 8.

### 4.4. CA Attention Mechanism

Considering the complexity of farm imagery, the CA attention mechanism [37] was added to the pyramid fusion network. This addition improves feature representation by introducing spatial coordinate information into the channel attention. The structure of the enhanced network is illustrated in Figure 9.

The coordinate attention (CA) mechanism encodes channel relationships and long-term dependencies in two steps: embedding coordinate information and generating coordinate attention. For the input, X encodes each channel with two spatial ranges of the pooled kernel (H,1) and (1,W) along the horizontal coordinates $x_{c}(h,i)$ and vertical coordinate $x_{c}(j,w)$. This can be expressed by Equations (4) and (5).
(4)$$z_{c}^{h}(h)=\frac{1}{W}\sum_{0\leq j<W}x_{c}(h,i)$$
(5)$$z_{c}^{w}(w)=\frac{1}{H}\sum_{0\leq j<H}x_{c}(j,w)$$

In this context, $z_{c}^{h}(h)$ and $z_{c}^{w}(w)$ are the feature encodings of the coordinate direction. After aggregating the features along these two spatial directions and performing dimensionality reduction, the fused directional feature information is obtained. This information is then extended by applying the sigmoid activation function to derive the attention weights. The final output can be expressed by Equation (6).
(6)$$y_{c}(i,j)=x_{c}(i,j)\times g_{c}^{h}(i)\times g_{c}^{w}(j)$$
where *g^h^* and *g^w^* are spatial direction weights. The coordinate attention (CA) mechanism incorporates coordinate information, significantly enhancing the ability of convolutional neural networks to capture long-range dependencies and extract detailed features.

In this study, we enhance the performance of the SparseInst network for segmenting farm elements by improving the multi-scale feature fusion capabilities of the neck network. This is achieved by adding a bottom-up aggregation path to form a new feature fusion network. Furthermore, we design a multi-scale attention module tailored to the characteristics of farm elements and integrate a coordinate attention mechanism to meet the segmentation requirements for farm elements.

### 4.5. Loss Function

The SparseInst network produces a fixed-size set of predictions. However, because the number of predictions may differ from the number of actual objects, accurately matching each prediction with a real object can be challenging. Therefore, the label assignment problem is formulated as a two-way matching issue using a matching score based on the Dice coefficient, denoted as C(i,k), which is used to measure the degree of matching between the *i* th prediction and the *k* th real object as in Equation (7).
(7)$$C(i,k)=p_{i,ck}^{1-a}\cdot DICE{\left(m_{i},t_{k}\right)}^{a}$$

*a* is a hyperparameter that balances the effects of the categorization score and the Dice coefficient. $P_{i,ck}$ is the probability that the *i* th prediction belongs to the *k* th true object category. $DICE(m_{i},t_{k})$ is the Dice coefficient of the *i* th prediction mask and the *k* th true mask, which is computed by Equation (8) as follows.
(8)$$DICE(m,t)=\frac{2\sum_{x,y}m_{xy}\cdot t_{xy}}{\sum_{x,y}m_{xy}^{2}+\sum_{x,y}t_{xy}^{2}}$$
where $m_{xy}$ and $t_{xy}$ represent the pixel values of the *i* th prediction mask and the *k* th true mask at position (x,y), respectively.

The training loss consists of the following components. Classification Loss employs Focal Loss [38] to deal with the class classification problem, especially to deal with class imbalance. Mask Loss evaluates the accuracy of the predicted mask in matching the true mask. Targeting Loss measures the probability of the target’s presence, using IoU-aware binary cross-entropy loss. The final training loss is defined as shown in Equation (9).
(9)$$L=\lambda_{c}\cdot L_{cls}+L_{mask}+\lambda_{s}\cdot L_{s}$$
where $\lambda_{c}$ and $\lambda_{s}$ are weighting factors.

## 5. Experimental Results and Analysis

### 5.1. Evaluation Indicators

Average precision (AP) [39] is selected as the evaluation metric for the instance segmentation model in the experiments. Precision reflects the model’s accuracy, while recall evaluates the model’s ability to recognize instances. Average precision, which represents the area under the precision–recall curve, is used to comprehensively assess the model’s performance on the test set and is calculated as shown in Equation (10).
(10)$$AP=\int_{0}^{1}Precision(Recall)d(Recall)$$

Precision and recall are calculated using Equations (11) and (12), respectively. The mean average precision (mAP) is the average of the *AP* values for all categories, as shown in Equation (13).
(11)$$Precision=\frac{TP}{TP+FP}$$
(12)$$Recall=\frac{TP}{TP+FN}$$
(13)$$\mathrm{mAP}=\frac{1}{N}\sum_{i=1}^{n}AP_{i}$$

*TP*, *FP*, and *FN* represent true positives, false positives, and false negatives under different IoU thresholds, respectively. IoU measures the degree of overlap between the predicted and true masks, defined as the ratio of the intersection to the union of the predicted and true mask regions. This experiment uses the average precision to evaluate the model’s performance across different IoU thresholds, ranging from 0.5 to 0.95. Frames Per Second (FPS) and cross-validation metrics are also used to evaluate model performance.

AP50: average precision when the IoU threshold is 0.5.AP75: average precision when the IoU threshold is 0.75.APs: average precision for small objects (area < 32^2^).APm: average precision for medium objects (32^2^ ≤ area < 96^2^).APl: average precision for large objects (area ≥ 96^2^).

### 5.2. Experimental Environment

The operating system used in this experiment is Linux 3.10 with an NVIDIA A100 GPU with 16 GB memory. The deep learning framework employed is PyTorch 1.10.0 and the parallel architecture used is CUDA 11.2. Due to the significant category difference between this dataset and the COCO dataset, the initial weights from the COCO pre-trained model were not utilized. The images for pre-training were resized to 1056 × 790 pixels. The initial learning rate was set to 0.00005, with a momentum [40] factor of 0.9 and a weight decay [41] of 0.05. The total number of iterations was 200,000, and the batch size was 2. Data enhancements such as scaling, horizontal flipping, and vertical flipping of images are added during the training process.

### 5.3. Ablation Study

#### 5.3.1. Comparison with Other Baselines

To verify the effectiveness of the improved model, comparisons were made with other segmentation algorithms, including the classical networks Mask R-CNN, Cascade–Mask [42], and the widely used SOLOv2 [43] and Condinst networks, based on mean average precision (mAP). In addition to these comparison experiments, cross-validation was performed to further assess the model’s generalization and robustness, ensuring that it did not overfit specific data. The network complexity (GFLOPs) and model inference speed (FPS) were also measured, with the results presented in Table 2.

The experimental results revealed that the segmentation accuracy of Mask R-CNN and Cascade–Mask networks is poor due to the large scale of the farm instance targets. Both networks utilize the RPN region proposal network for target detection, generating nine candidate regions with aspect ratios of 1:1, 1:2, and 2:1, and then segmenting the instance masks based on these regions. However, the complex structure and scale of farm targets make it difficult to generate accurate candidate regions, resulting in imprecise segmentation. In contrast, SOLOv2, CondInst, and SparseInst are single-stage anchor-free algorithms, which avoid the limitations of RPN-based detection. The results indicate that the improved SparseInst increases the average precision by 10.8 and 12.8 percentage points compared to these two algorithms. The cross-validation results further confirm that the model demonstrates strong generalization and robustness, performing well on five randomly divided datasets and outperforming other single-stage anchor-free networks. It also has the fastest inference speed and the lowest network complexity, making SparseInst the optimal choice for instance segmentation of farm images.

#### 5.3.2. Ablation Study Results and Analysis

To validate the effectiveness of each improvement strategy, this study used SparseInst as the benchmark network. Ablation experiments were conducted to assess each enhancement, and the results are presented in Table 3.

The ablation experiment, which focused on a farm aerial survey scene, revealed that incorporating the MSA module into the feature fusion network enhances the average precision by 1.32 percentage points compared to the baseline network, and the AP75/% increases by 2.1 percentage points. In terms of average precision across different pixel areas, both the baseline network and the network with the MSA module show limited performance in distinguishing small targets (less than 32 × 32 pixels) and medium targets (between 96 × 96 pixels). However, the APl/% for large targets improved by 1.927 percentage points. These results demonstrate that the proposed MSA module, which utilizes atrous convolution, significantly benefits the segmentation of large-scale targets. The addition of an aggregation path to the feature fusion structure enhances the integration of complex multi-target features based on the MSA module, particularly improving the handling of narrow and variable targets, such as mechanized roads and highways. The incorporation of the CA module significantly enhances the accuracy of the overall network structure. The average accuracy improves by 2.103 percentage points compared to the baseline model. Additionally, the average precision at IoU thresholds of 50% (AP50) and 75% (AP75) increased by 0.708 and 3.167 percentage points, respectively. For large targets exceeding 96 × 96 pixels, the improvement over the baseline network is 2.69 percentage points. Overall, the enhanced SparseInst-based network structure demonstrates a significant improvement in average accuracy. However, both the baseline and improved networks continue to face challenges in detecting small-scale targets, such as outletpile.

Table 4 shows that this is the average AP/% accuracy of the improved network for the farm categories, with eight categories. Most of these categories consist of plotlands and adjacent forest belts, which represent the majority of the categories and exhibit the largest scales. The dataset also includes arable roads (paths between plots) and irrigation watercourses. Due to the UAV aerial imagery, these structures often span entire images, presenting narrow and irregular shapes with colors similar to the background, which makes segmentation challenging and tends to result in inaccuracies.

The results show that the highest accuracy of plotland and river reaches is more than 90%. The MSA module significantly improves the overall model accuracy, particularly enhancing the extraction capabilities for targets with complex and irregular structures, such as forest belts, arable roads, wild grasslands, watercourses, and rivers. This improvement demonstrates that densely connected atrous convolutions effectively capture spatial information across multiple scales. Notably, the largest gains are observed in the segmentation of watercourses and rivers, with accuracy improvements of 2.506 and 3.581 percentage points, respectively. The addition of an aggregation path to the feature fusion structure significantly enhances the segmentation performance for roads and mechanized roads, with improvements of 3.197 and 7.385 percentage points, respectively, compared to the baseline network. This modification also enhances information transfer and fusion, particularly for arable roads, which exhibit large variations in structural scales and complexity. Incorporating the CA module further enhances the network’s feature extraction capability. However, both the improved and baseline networks struggle with the accurate segmentation of very small-scale targets, such as outletpiles. This challenge arises because outlet piles, distributed within plots, have small shapes and colors that blend into the surrounding land, making them difficult to distinguish and segment effectively.

#### 5.3.3. Visual Analysis of Experimental Outcomes

As shown in Figure 10, the segmentation results of the improved strategy are compared with those of the other two models. Due to the inapplicability of Mask R-CNN and Cascade–Mask to farm image segmentation, their results are low due to accuracy. The six images include categories such as plotlands, forest belts, highways, watercourses, rivers, arable roads, and highways. The results demonstrate that both SparseInst and the improved SparseInst achieve high accuracy in segmenting plotlands, highways, and rivers. Notably, the original SparseInst missed the detection and segmentation of forest belts, whereas the improved SparseInst achieved more comprehensive segmentation of forest belts, wild grasslands, and arable roads.

In order to verify the network’s performance on real-world images, the final trained model was used to recognize high-resolution farm images, as shown in Figure 11. Since the image resolution is 5280 × 3956, the available computational resources could not process such large-resolution images, so the network was trained using uniformly cropped images with a resolution of 1760 × 1319. As a result, the model demonstrated excellent segmentation performance on the cropped images but incomplete segmentation for targets in the uncropped high-resolution images. However, the improved SparseInst still outperforms the original SparseInst network for high-resolution farm image segmentation. This issue arises because the feature maps generated from low-resolution images during training are specific to certain scales and may not directly align when applied to high-resolution images, thus affecting the model’s segmentation accuracy.

In fact, in performing the construction of the farm layer, a complete map is stitched together from numerous small maps. The operation of this study for cropping the training image does not affect the construction of subsequent layers. However, to address this deficiency, plans could include image scaling, where the original image is cropped as close to its original resolution as possible, and image scaling is used to keep the image undistorted. Additionally, implementing multi-scale training using images of different resolutions during the training process can improve the model’s ability to generalize to images of varying scales.

#### 5.3.4. HRSID Validation

For the segmentation of high-resolution farm remote sensing images, this study adopts the SparseInst single-stage anchorless instance segmentation algorithm. To further validate the effectiveness of the proposed approach, a comparative validation was conducted using the High-Resolution SAR Images Dataset (HRSID) with the same single-stage anchorless instance segmentation algorithm. The HRSID is a high-resolution synthetic aperture radar (SAR) remote sensing dataset for vessel detection, consisting of 5604 high-resolution SAR images. Since the SAR images are grayscale, with vessels and backgrounds distinguishable in black and white, it is easy to evaluate the accuracy of the network’s segmentation masks. Given the similarity between the HRSID and the UAV farm dataset in this study, both falling within the same category of remote sensing, the HRSID is suitable for validating the effectiveness of the improved method presented in this research, and the results are presented in Table 5.

The results indicate that the improved SparseInst network achieves the highest average accuracy and has the lowest complexity compared to the SOLOv2 and CondInst networks. The improved network increases the complexity compared to the baseline network but effectively enhances model accuracy. The segmentation results for vessels are illustrated in Figure 10. Validation experiments with the HRSID confirm that the algorithm performs effectively in single-stage anchorless instance segmentation and the model’s ability to generalize. The segmentation results are shown in Figure 12.

## 6. Conclusions

This paper focuses on UAV aerial survey images of farms, constructing a dedicated farm aerial survey image dataset. It proposes an instance segmentation algorithm with an improved SparseInst network, specifically designed to address the challenges of complex target structures and significant scale variations in farm images. We design the MSA module, introduce the CA module, and enhance the neck network. Based on ablation experiments and comparisons with other classical networks, the following conclusions are drawn.

(1) The experimental results on the self-constructed farm imagery dataset demonstrate that the improved SparseInst enhances the average accuracy by 2.103, 0.708, and 3.167 percentage points, respectively, compared to the baseline model. Notably, the proposed MSA has significantly improved the detection and segmentation of watercourses and river targets in complex scenes. The arable road is one of the most complex elements due to its color similarity to the surrounding background. However, the improved feature fusion network effectively enhances the perception of such challenging semantic information, enabling better recognition of arable roads.

(2) Comparison experiments with the Mask R-CNN, Cascade–Mask, SOLOv2, and Condinst algorithms demonstrated that the improved SparseInst outperforms SOLOv2 and Condinst by 10.8 and 12.8 percentage points, respectively. The experiments on the farm aerial survey dataset conclude that instance segmentation algorithms based on anchor-free methods are better suited for handling large-scale targets compared to those built on object detection frameworks.

The enhanced accuracy of the model presented in this paper meets the requirements for constructing farm layers, thereby improving manual construction efficiency and reducing labor costs. Additionally, it provides technical support for instance segmentation in UAV imagery.

Despite significant advancements in UAV farm image segmentation, current datasets still face limitations, particularly in the number of small target instances and the ability to segment targets with complex structures. Future research should focus on increasing the instances of small-scale and complex-structured targets and expanding the dataset to include a wider range of farm categories. Due to the model’s poor inference performance on high-resolution images, the next step is to create a multi-scale farm image dataset to improve the model’s segmentation performance on such images. Additionally, optimizing model parameters is crucial for enhancing both accuracy and generalization capability.

## Figures and Tables

**Figure 1 sensors-24-05990-f001:**
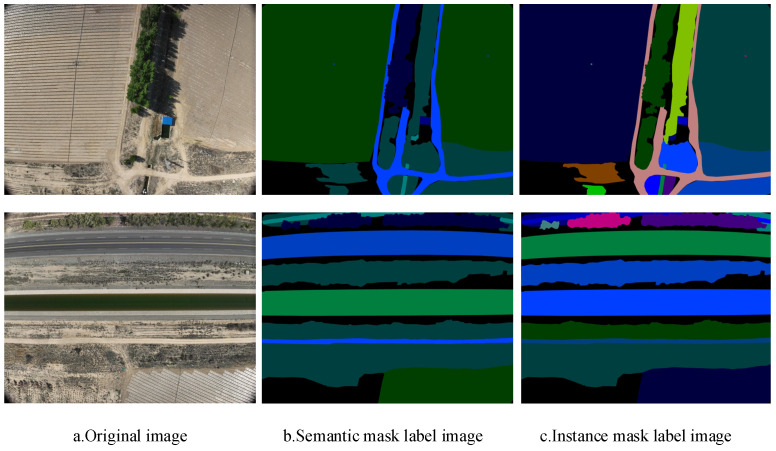
Farm scene mask image.

**Figure 2 sensors-24-05990-f002:**
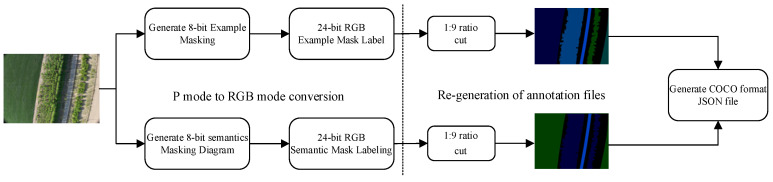
Data processing flowchart.

**Figure 3 sensors-24-05990-f003:**
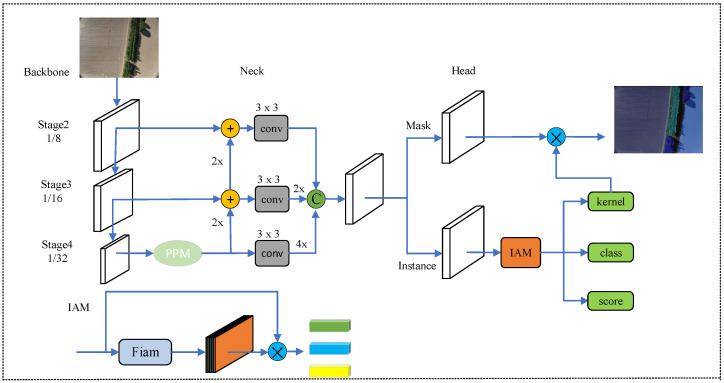
SparseInst network architecture. The SparseInst network architecture comprises three main components: the backbone, the encoder, and the IAM-based decoder. The backbone extracts multi-scale image features from the input image, specifically {stage2, stage3, stage4}. The encoder uses a pyramid pooling module (PPM) [30] to expand the receptive field and integrate the multi-scale features. The notation ‘4×’ or ‘2×’ indicates upsampling by a factor of 4 or 2, respectively. The IAM-based decoder is divided into two branches: the instance branch and the mask branch. The instance branch utilizes the ‘IAM’ module to predict instance activation maps (shown in the right column), which are used to extract instance features for recognition and mask generation. The mask branch provides mask features M, which are combined with the predicted kernels to produce segmentation masks.

**Figure 4 sensors-24-05990-f004:**
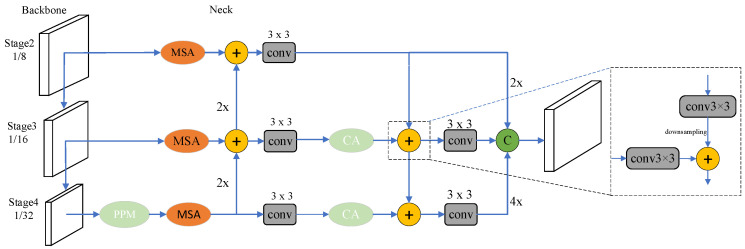
Improved SparseInst neck network PPM refers to the pyramid pooling module, MSA refers to the multi-scale attention module, 2× and 4× denote upsampling by a factor of 2 and 4, respectively, 3 × 3 denotes a convolution operation with a kernel size of 3, + denotes element-wise summation, and CA refers to the coordinate attention module.

**Figure 5 sensors-24-05990-f005:**
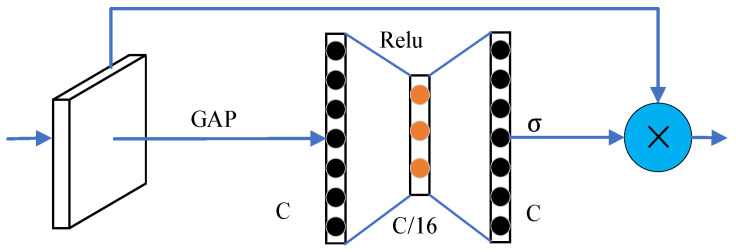
Channel attention mechanism. GAP stands for global average pooling, relu is the rectified linear unit activation function, σ represents the Sigmoid activation function, C denotes the number of channels, and × denotes element-wise multiplication.

**Figure 6 sensors-24-05990-f006:**
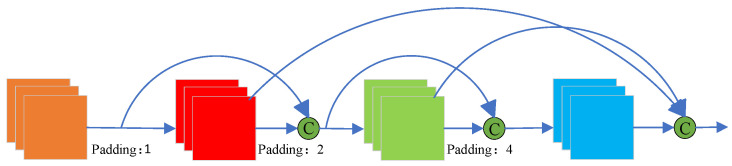
Dense connection diagram padding refers to the dilation rate of the convolution kernel, and C denotes feature concatenation.

**Figure 7 sensors-24-05990-f007:**
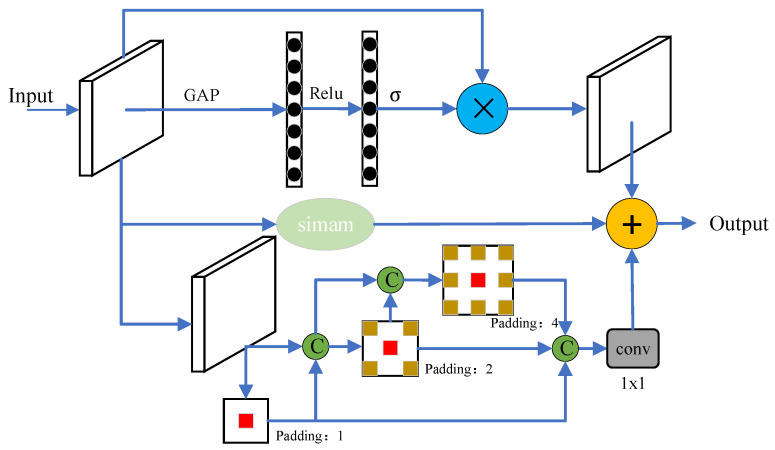
Multi-scale attention module (MSA). GAP stands for global average pooling, relu is the rectified linear unit activation function, *σ* represents the activation function, padding refers to the dilation rate coefficient, and c denotes concatenation. + is element-by-element addition. × is a matrix product.

**Figure 8 sensors-24-05990-f008:**
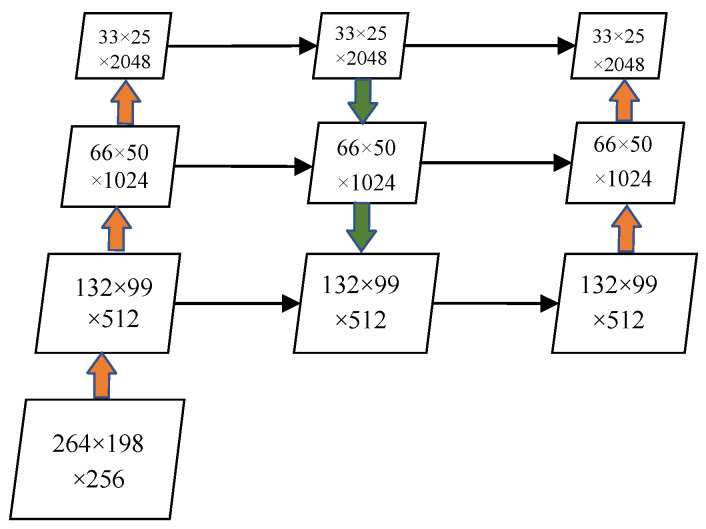
PADPN network architecture.

**Figure 9 sensors-24-05990-f009:**
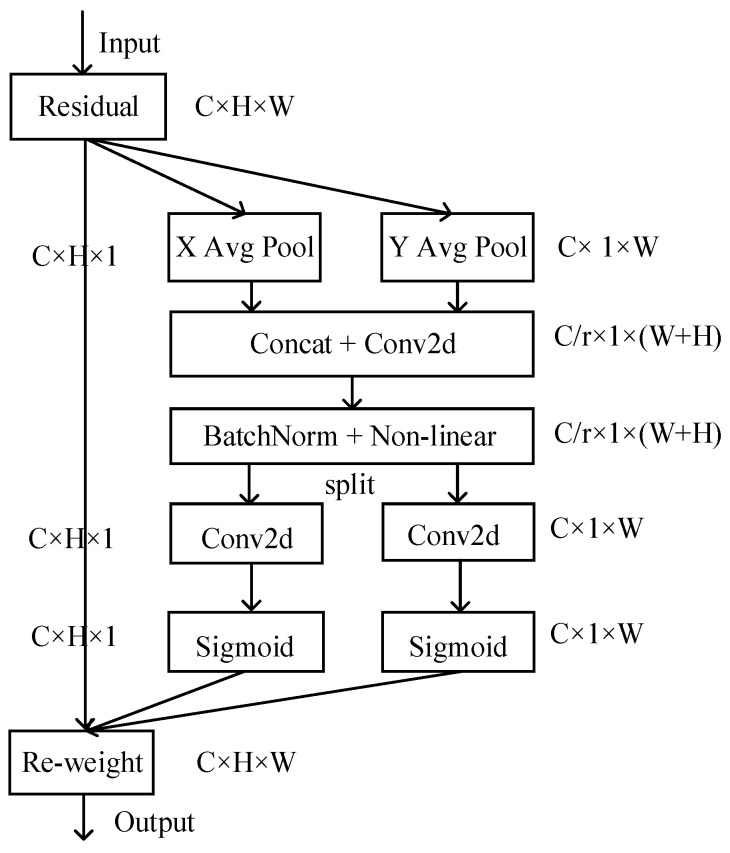
Coordinate attention blocks X Y (avg pool) denote global pooling along the h and w directions, BatchNorm refers to batch normalization, non-linear represents the non-linear activation function, split denotes splitting along the channel dimension, and Sigmoid represents the activation function.

**Figure 10 sensors-24-05990-f010:**
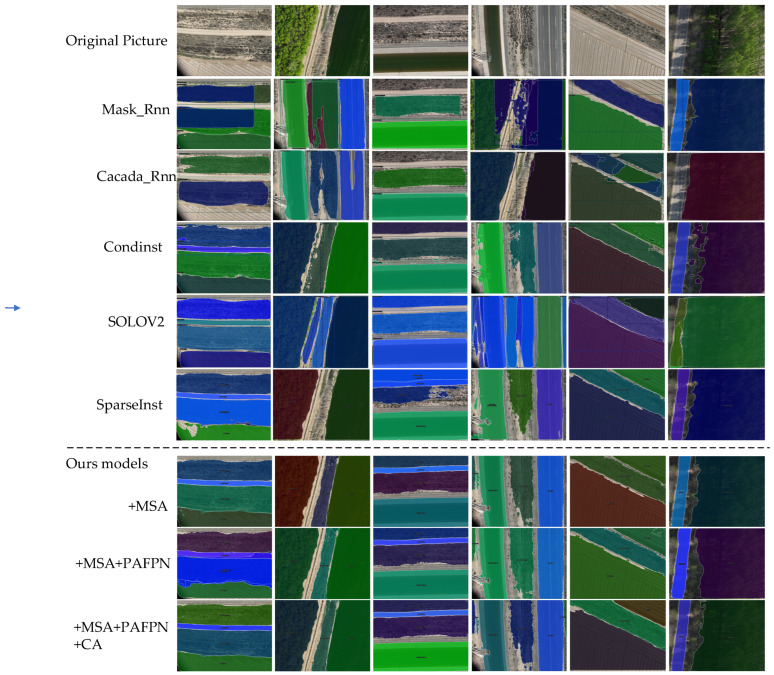
Visualization results.

**Figure 11 sensors-24-05990-f011:**
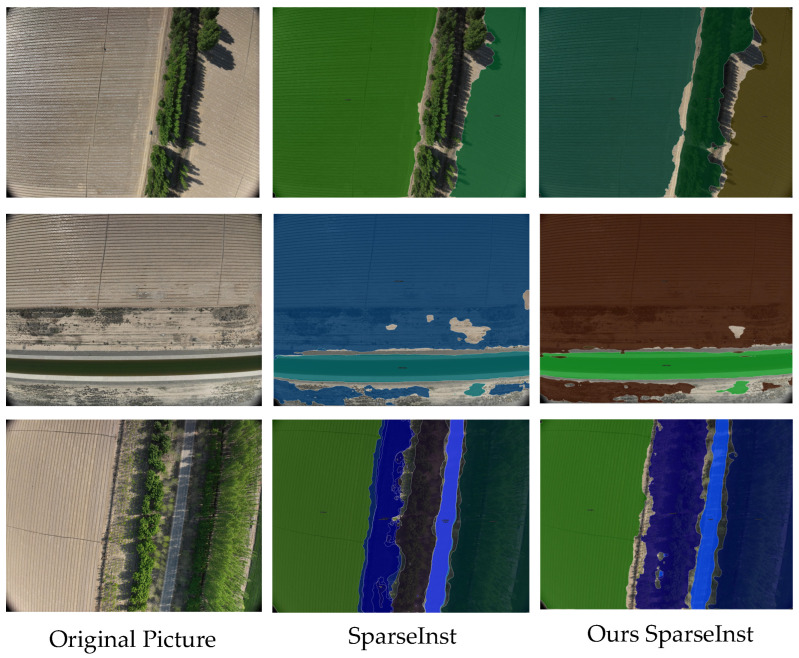
High-resolution image visualization results.

**Figure 12 sensors-24-05990-f012:**
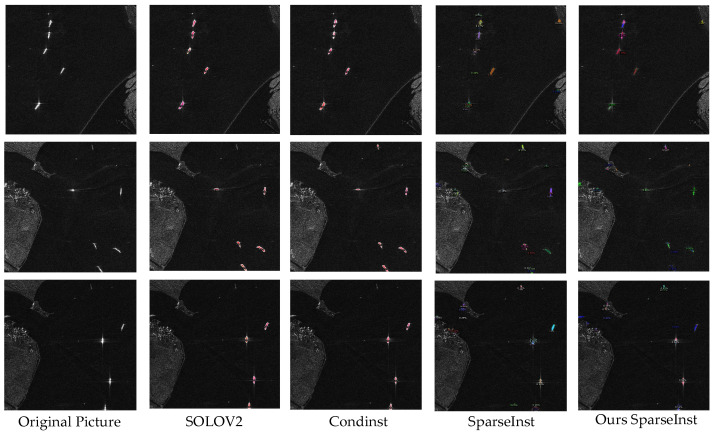
HRSID visualization results.

**Table 1 sensors-24-05990-t001:** The distribution of the number of elements in the farm imagery dataset.

Plotland	Forest_Belt	Outletpile	Arable Road	Wild_Grasland	River	Highway	Watercourses	Total
9465	2852	1689	2244	5493	491	978	968	24,180

**Table 2 sensors-24-05990-t002:** Comparison with other algorithms.

Model	mAP/%	AP50/%	AP75/%	APs/%	APm/%	APl/%	GFLOPs	CV (mAP)	FPS
Mask R-CNN	31.1	19.6	9.4	0	0	12.5	226	29.0	23.1
Cascade–Mask	39.0	54.0	39.4	0.3	2.0	45.3	1749	33.2	12.7
SOLOv2	51.0	66.7	52.5	0.7	9.8	59.5	224	43.8	27.5
Condinst	49.0	62.8	50.7	2.7	5.3	57.2	178	24.3	21.6
Improving SparseInst	61.8	80.9	62.6	3.2	6.0	70.7	157	57.3	38.8

Note: mAP/% represents mean accuracy, AP50/% is average precision at an IoU threshold of 50%, AP75/% is average precision at an IoU threshold of 75%. APs/% is average precision for small objects, APm/% is for medium objects, APl/% is average precision for large objects. CV refers to the mean accuracy obtained from cross-validation.

**Table 3 sensors-24-05990-t003:** Ablation study results for six types of mean average precision (mAP).

MSA	PAFPN	CV	mAP/%	AP50/%	AP75/%	Aps/%	APm/%	APl/%
×	×	×	59.751	80.227	59.465	3.585	4.340	68.057
√	×	×	61.071	79.991	61.565	3.028	8.105	69.984
√	√	×	61.050	79.443	62.421	3.340	4.735	69.882
√	√	√	61.854	80.935	62.632	3.264	6.008	70.747

Note: ‘×’ indicates that the corresponding strategy is not used, ‘√’ indicates that the corresponding strategy is adopted.

**Table 4 sensors-24-05990-t004:** Mean average precision of farm categories under different improvement strategies (AP/%).

Category	Original SparseInst	+MSA	+MSA + PAFPN	+MSA + PAFPN + CA
Plotland	93.464	94.205	94.382	94.587
Forest_Belt	62.705	65.743	64.831	66.861
Outletpile	16.541	14.089	15.690	15.741
Highway	82.038	82.097	85.235	82.872
Arable Road	41.045	45.709	48.430	47.844
Wild_Grassland	30.697	34.119	32.912	36.338
Watercourse	52.532	55.038	50.105	53.692
River	93.987	97.568	96.816	96.900

**Table 5 sensors-24-05990-t005:** Comparative validation of the HRSIDs.

Model	mAP/%	AP50/%	AP75/%	APs/%	APm/%	APl/%	GFLOPs	FPS
SOLOv2	27.2	57.0	22.2	26.5	34.5	2.5	178	19.8
Condinst	31.9	65.8	28.9	31.5	38.9	6.7	210	11.2
SparseInst	31.4	55.8	35.1	31.1	37.3	7.6	118	26.0
Improving SparseInst	34.5	61.0	38.7	34.2	40.4	6.3	139	22.7

## Data Availability

The data supporting the findings of this study are available upon request from the readers.

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
