# Peer review of "An Improved Instance Segmentation Method for Complex Elements of Farm UAV Aerial Survey Images"

_sensors, 2024, doi:10.3390/s24185990_

Round 1
Reviewer 1 Report
Comments and Suggestions for Authors
Constructing farm aerial survey layers facilitates unmanned farm machine operations, including path planning and warning. To address the inefficiencies and high costs associated with traditional layer construction, this study proposes a high-precision instance segmentation algorithm based on SparseInst. However many technologies you used are proposed by other researchers. It seems that you just combined the work of others. Please clarify the novelty and contributions of your paper in detail. In order to make the experimental results convincing, could you please show the cross-validation results and do some statistical tests?
Comments on the Quality of English Language1) abstract line 10, a high not an high
Author Response
Dear Reviewers:
Thank you for your letter and for the reviewers’ comments concerning our manuscript entitled “An Improved Instance Segmentation Method for Complex Elements of Farm UAV Aerial Survey Images” (ID: sensors-3184367). Those comments are all valuable and very helpful for revising and improving our paper, as well as the important guiding significance to our researches. We have studied comments carefully and have made correction which we hope meet with approval.
SUGGESTIONS FROM EDITOR
- Constructing farm aerial survey layers facilitates unmanned farm machine operations, including path planning and warning. To address the inefficiencies and high costs associated with traditional layer construction, this study proposes a high-precision instance segmentation algorithm based on SparseInst. However many technologies you used are proposed by other researchers. It seems that you just combined the work of others. Please clarify the novelty and contributions of your paper in detail.
The author’s answer
Thank you for your comments. The challenge in this study lies in the wide coverage of farm images, where natural features are prominent, and the targets to be segmented (mechanized roads, watercourse, and heathland) exhibit complex characteristics and significant scale variations. Traditional instance segmentation networks struggle to segment these contours effectively. Therefore, the novelty of this study is the improvement of the SparseInst network to better address the specific characteristics of farm targets.
A multi-scale attention module (MSA) is proposed to address the characteristics of farm images, effectively improving the segmentation accuracy of targets such as watercourses and forest belts; Since the original network struggles to enhance the feature extraction for small targets, this study introduces a bottom-up aggregation path to form a new feature fusion network and integrates the CA module to further strengthen the network's extraction capabilities.
In the field of instance segmentation, few studies have addressed high-resolution images, such as UAV farm aerial surveys, where the segmented targets exhibit complex structures and significant scale variations within the same class. To address this, it took several months to complete the production of a large UAV farm dataset. The proposed segmentation algorithm not only meets the requirements for constructing maps from this dataset but also provides technical support for the segmentation of other high-resolution UAV images.
- In order to make the experimental results convincing, could you please show the cross-validation results and do some statistical tests?
The author’s answer
Thank you very much for your professional advice, to make the results more convincing, cross-validation(CV) was done for each comparison model as requested, as well as counting the FPS speeds for each model. The results are shown in Table 3 of 5.3.2. Ablation Study results and analysis.
- abstract line 10, a high not an high
The author’s answer
Thank you for your comments To address the inefficiencies and high costs associated with traditional layer construction, this study proposes a high-precision instance segmentation algorithm based on SparseInst.
The full text was rechecked for logical and grammatical issues, with all changes marked in red
Once again, thank you very much for your comments and suggestions. And we hope that the revised manuscript can be accepted by [sensors].
Thank you and best regards.

Reviewer 2 Report
Comments and Suggestions for Authors
11. Grammatical structure of some sentences should be reviewed
Example
· Constructing farm aerial survey layers facilitates unmanned farm machine operations, 8 including path planning and warning
· This technology offers significant benefits by monitoring and decision-making support for unmanned farm machinery on large farms.
2. And add descriptions of the organization of this paper at the end of the Introduction.
3. Extensive research under the introduction should be detached form introduction and place under a new section named Related Works or Previous research. Then the scope can be extended to include more related works
4. Other evaluation metrics like the Mean Average Precision (mAP) or the Panoptic Quality could have been used
5. A single dataset was used. The proposed model could have been tested on other datasets which may limit the model's ability to generalize
6. Time complexity of the model should have been tested
Comments on the Quality of English Language1. Grammatical structure of some sentences should be reviewed
Example
· Constructing farm aerial survey layers facilitates unmanned farm machine operations, 8 including path planning and warning
· This technology offers significant benefits by monitoring and decision-making support for unmanned farm machinery on large farms.
Author Response
Dear Reviewers:
Thank you for your letter and for the reviewers’ comments concerning our manuscript entitled “An Improved Instance Segmentation Method for Complex Elements of Farm UAV Aerial Survey Images” (ID: sensors-3184367). Those comments are all valuable and very helpful for revising and improving our paper, as well as the important guiding significance to our researches. We have studied comments carefully and have made correction which we hope meet with approval.
SUGGESTIONS FROM EDITOR
- Grammatical structure of some sentences should be reviewed
Example
- Constructing farm aerial survey layers facilitates unmanned farm machine operations, 8 including path planning and warning
- This technology offers significant benefits by monitoring and decision-making support for unmanned farm machinery on large farms.
The author’s answer
Thank you very much for your professional advice. Here is the revised sentence: Farm aerial survey layers can assist in unmanned farm operations, such as planning paths and early warnings.
In response to your second sentence: the language has been reorganized in the first paragraph of the introduction to provide more detail on the relevance of this study, marked in red.
The full text was rechecked for logical and grammatical issues, with all changes marked in red
- And add descriptions of the organization of this paper at the end of the Introduction.
The author’s answer
Thank you very much for your comments. A description of the organization of the paper has been added and marked in red at the end of the introduction. The section is shown below:
The rest of the paper is organized as follows. Section 2 presents research results on instance segmentation in the field of UAVs. Section 3 describes collecting and constructing the farm aerial survey dataset. Section 4 details the SparseInst network and the improvement methods, including the multi-scale attention mechanism, aggregation path, and CA attention module, tailored to the characteristics of farm target structures. Section 5 analyzes the experimental results and includes the design and execution of comparison, ablation, and validation experiments on the HRSID dataset. The conclusions are presented in Section 6.
- Extensive research under the introduction should be detached form introduction and place under a new section named Related Works or Previous research. Then the scope can be extended to include more related works
The author’s answer
Thank you for your valuable suggestions. The related research has been included as a separate chapter, “2. Related Works.” Some relevant studies have been added as requested and are highlighted in red.
- Other evaluation metrics like the Mean Average Precision (mAP) or the Panoptic Quality could have been used
The author’s answer
Thank you very much for your professional advice. Forgive the lack of detail in the presentation of the evaluation indicators, which have been reorganized in Evaluation Indicators 5.1. mAP is the average of the AP values for all categories. The first evaluation indicator in Table 2, Table 3, and Table 5 is mAP. To make the experimental results more convincing, this modification adds the evaluation indicators of cross-validated experimental CV (mAP) and time-complexity FPS, which can be seen in Table 2.
Panoptic Quality; Most of the panoptic segmentation evaluation methods are different from the example segmentation methods used for the farm images in this study. Additionally, Panoptic Quality requires the computation of IoU (Intersection over Union), which some of the compared networks in this paper do not support, making it challenging to use this metric.
We refer to several kinds of literature in the field of panoptic segmentation and instance segmentation in the meantime, and the evaluation metrics they use can corroborate this. Panoptic segmentation:
Jaus, Alexander, Kailun Yang, and Rainer Stiefelhagen. "Panoramic panoptic segmentation: Insights into surrounding parsing for mobile agents via unsupervised contrastive learning." IEEE Transactions on Intelligent Transportation Systems 24.4 (2023): 4438-4453.
Kim, Dahun, et al. "Video panoptic segmentation." Proceedings of the IEEE/CVF Conference on Computer Vision and Pattern Recognition. 2020.
- A single dataset was used. The proposed model could have been tested on other datasets which may limit the model's ability to generalize
The author’s answer
Thank you very much for your professional advice. Following your suggestion, we used the HRSID remotely sensed vessel dataset to test the generalization capabilities of the model and performed an inference visualization. This is detailed in 5.3.4 HRSID Dataset Validation, where the results demonstrate that the improved model presented in this paper exhibits significant advantages when applied to other datasets. This section has been highlighted in red.
- Time complexity of the model should have been tested
The author’s answer
Thanks to your suggestion, the time complexity (FPS) of both the improved model presented in this paper and the comparison models has been tested, as shown in Tables 2 and 5.
Once again, thank you very much for your comments and suggestions. And we hope that the revised manuscript can be accepted by [sensors].
Thank you and best regards.

Reviewer 3 Report
Comments and Suggestions for Authors
Overall comment: While the improvements to the method do not significantly surpass the original network, the structure and writing of the paper are quite clear. Therefore, I have only a few minor suggestions.
Detailed comments:
- I recommend that the authors provide a description of the structure of the Multi-Scale Attention module as shown in Figure 7, before delving into too many detailed specifics.
- The first time an abbreviation is used in the text, it should be written out in full. Please recheck to ensure this is consistently applied.
- I suggest adding a comparative experiment where the trained network is used to segment the original, uncropped images to validate the network's performance on real-world imagery, comparing it with classic networks like SparseInst.
- We know that the parameter settings of a network can impact its architecture. Was data augmentation used during the training process?
Author Response
Dear Reviewers:
Thank you for your letter and for the reviewers’ comments concerning our manuscript entitled “An Improved Instance Segmentation Method for Complex Elements of Farm UAV Aerial Survey Images” (ID: sensors-3184367). Those comments are all valuable and very helpful for revising and improving our paper, as well as the important guiding significance to our researches. We have studied comments carefully and have made correction which we hope meet with approval.
SUGGESTIONS FROM EDITOR
- I recommend that the authors provide a description of the structure of the Multi-Scale Attention module as shown in Figure 7, before delving into too many detailed specifics.
The author’s answer
Thank you very much for your professional advice, the description of Figure 7 in this paper has been clarified, and the language has been reorganized to provide a more detailed explanation of the network structure depicted in Figure 7. This part has been marked in red. The section is shown below:
The structure of the MSA network designed for this experiment is illustrated in Figure 7. The fully connected layer is replaced by two 11 convolutions to obtain channel semantic information weights, which are then matrix-multiplied with the input features to produce the enhanced channel semantic information weighted map. Another branch applies three parallel atrous convolutions for large-scale feature extraction. However, this parallel design does not optimally utilize multi-scale semantic information and shares limitations similar to the feature pyramid network (FPN), often resulting in an incomplete representation of small-scale semantic information in the feature maps, leading to potential loss. To address this, a dense connection structure is introduced between the three atrous convolutions, as depicted in Figure 6. This design fuses small-scale and complex structural semantic information through channel concatenation, ensuring the retention of small-scale semantic information while extracting large-scale targets. Following a 11 dimensionality reduction, the output is summed with the channel information weighted map and the features enhanced by the Simam mechanism to generate the multiscale semantic feature map. The results demonstrate that this Multiscale Attention Module (MSA) significantly enhances the extraction of large-scale target features within the farm environment.
Additionally, the logical structure and grammatical issues throughout the article have been revised and reorganized. The changes have been highlighted in red.
- The first time an abbreviation is used in the text, it should be written out in full. Please recheck to ensure this is consistently applied.
The author’s answer
Thank you very much for your comments. The problem with full names has been corrected in the text, and all full names that have been changed have been red-flagged, for example:
multiscale attention module (MSA)
region proposal network (RPN)
path aggregation network (PAFPN)
- I suggest adding a comparative experiment where the trained network is used to segment the original, uncropped images to validate the network's performance on real-world imagery, comparing it with classic networks like SparseInst.
The author’s answer
Thank you very much for your professional advice. To evaluate the network's performance on real-world images, the final trained model was applied to recognize high-resolution farm images, with the results shown in Figure 11. Due to the large image resolution, some categories were not accurately segmented, as the low-resolution images used in training generate scale-specific feature maps, which may not match well during inference on high-resolution images, thus impacting segmentation accuracy.
In practice, the construction of farm layers involves stitching together a complete map from numerous smaller maps, meaning the image cropping during training does not affect the final layer construction. However, to address this limitation, future work will include the development of multi-scale datasets and image downscaling techniques to enhance the model's generalization to high-resolution images.
- We know that the parameter settings of a network can impact its architecture. Was data augmentation used during the training process?
The author’s answer
We use data augmentation during the training process, including scaling, horizontal flipping, and vertical flipping of the images. Future work will incorporate additional data enhancement techniques, such as adding noise, color transformation, and blurring, to evaluate the impact of these enhancements on the model’s performance with farm images.
Once again, thank you very much for your comments and suggestions. And we hope that the revised manuscript can be accepted by [sensors].
Thank you and best regards.

Round 2
Reviewer 1 Report
Comments and Suggestions for Authors
Recommend to accept
Reviewer 2 Report
Comments and Suggestions for Authors
The comments have been addresses in the revised manuscript
